# Distinct Cytosolic Complexes Containing the Type III Secretion System ATPase Resolved by Three-Dimensional Single-Molecule Tracking in Live *Yersinia enterocolitica*

Joshua R. Prindle,[a] Yibo Wang,[a] Julian M. Rocha,[a] Andreas Diepold,[b] (ID) Andreas Gahlmann[a,c]

[a]Department of Chemistry, University of Virginia, Charlottesville, Virginia, USA
[b]Max Planck Institute for Terrestrial Microbiology, Marburg, Germany
[c]Department of Molecular Physiology & Biological Physics, University of Virginia School of Medicine, Charlottesville, Virginia, USA

**ABSTRACT** The membrane-embedded injectisome, the structural component of the virulence-associated type III secretion system (T3SS), is used by Gram-negative bacterial pathogens to inject species-specific effector proteins into eukaryotic host cells. The cytosolic injectisome proteins are required for export of effectors and display both stationary, injectisome-bound populations and freely diffusing cytosolic populations. How the cytosolic injectisome proteins interact with each other in the cytosol and associate with membrane-embedded injectisomes remains unclear. Here, we utilized three-dimensional (3D) single-molecule tracking to resolve distinct cytosolic complexes of injectisome proteins in living *Yersinia enterocolitica* cells. Tracking of the enhanced yellow fluorescent protein (eYFP)-labeled ATPase *Ye*SctN and its regulator, *Ye*SctL, revealed that these proteins form a cytosolic complex with each other and then further with *Ye*SctQ. *Ye*SctNL and *Ye*SctNLQ complexes can be observed both in wild-type cells and in Δ*sctD* mutants, which cannot assemble injectisomes. In Δ*sctQ* mutants, the relative abundance of the *Ye*SctNL complex is considerably increased. These data indicate that distinct cytosolic complexes of injectisome proteins can form prior to injectisome binding, which has important implications for how injectisomes are functionally regulated.

**IMPORTANCE** Injectisomes are membrane-embedded, multiprotein assemblies used by bacterial pathogens to inject virulent effector proteins into eukaryotic host cells. Protein secretion is regulated by cytosolic proteins that dynamically bind and unbind at injectisomes. However, how these regulatory proteins interact with each other remains unknown. By measuring the diffusion rates of single molecules in living cells, we show that cytosolic injectisome proteins form distinct oligomeric complexes with each other prior to binding to injectisomes. We additionally identify the molecular compositions of these complexes and quantify their relative abundances. Quantifying to what extent cytosolic proteins exist as part of larger complexes in living cells has important implications for deciphering the complexity of biomolecular mechanisms. The results and methods reported here are thus relevant for advancing our understanding of how injectisomes and related multiprotein assemblies, such as bacterial flagellar motors, are functionally regulated.

**KEYWORDS** biophysics, physiology, protein-protein interactions, secretion systems, single-molecule tracking

Address correspondence to Andreas Gahlmann, agahlmann@virginia.edu.

The authors declare no conflict of interest.

**B**acterial type III secretion systems (T3SS) are used by Gram-negative bacteria to assemble flagellar motors for cell motility and injectisomes that translocate virulence factors, called effector proteins, into eukaryotic host cells. Injectisomes are expressed by many bacterial pathogens, including *Escherichia coli*, *Salmonella*, *Pseudomonas*, *Shigella*, and *Yersinia*, which are responsible for widespread human disease both historically and currently. Flagellar motors and injectisomes are large, multicomponent protein structures that span two and sometimes

three cell membranes, as well as the bacterial cell wall (1). Assembly of the injectisome is achieved through hierarchal secretion of early, intermediate, and late secretion substrates (1–7). Early secretion substrates form an extracellular needle that extends away from the bacterial cell envelope. Upon contact with a host cell, middle secretion substrates are secreted through the needle to form a translocon pore in the eukaryotic cell membrane (1–4). Late secretion substrates, also called effector proteins, are then translocated into the host cell cytosol (5–7). While there are species-specific differences among the effector proteins, the structural proteins of injectisomes are highly conserved (8).

Molecular-resolution structures of the fully assembled cytosolic injectisome complex remain elusive. The cytosolic injectisome proteins SctN, -L, -Q, and -K associate with the injectisome loosely and transiently, so that structural imaging of fully assembled injectisomes is possible *in situ* only through the use of cryo-electron tomography (cryo-ET) (9–13). Injectisomes in *Salmonella* and *Shigella* minicell mutants were found to not exhibit continuous densities representative of a cytosolic ring (C ring) (9, 10), a feature that has been observed consistently for bacterial flagellar motors (3, 14). Instead, the subtomogram averages suggested six distinct pods composed of SctK, -Q, and -L that seem to cradle the hexameric ATPase (9, 10). More recently, cryo-focused ion beam milling followed by cryo-ET enabled the visualization of *Yersinia enterocolitica* injectisomes inside the phagosomes of infected human myeloid cells (15). In contrast to the six distinct cytosolic pods observed in *Shigella* and *Salmonella* minicell mutants, the *Y. enterocolitica* injectisome contained more continuous, ring-like densities likely composed of SctD, -K, and -Q and a 6-fold symmetric, cradle-like structure likely composed of SctL.

The cytosolic injectisome proteins have substantial freely diffusing populations. *In vivo* fluorescence recovery after photobleaching (FRAP) experiments revealed a continual exchange (on a 1-min time scale) between freely diffusing and injectisome-bound states for *Y. enterocolitica* SctQ (*Ye*SctQ). Interestingly, the *Ye*SctQ exchange rate at the injectisome increased by a factor of 2 upon chemical activation of secretion (16), suggesting that structural dynamism may be important for functional regulation of type 3 secretion. Fluorescence correlation spectroscopy (FCS) further revealed the presence of freely diffusing and injectisome-bound populations for *Ye*SctK, -L, and -N (17). In that study, the average diffusion rate for each cytosolic injectisome protein shifted upon chemical activation of secretion. These results further point to a dynamic and adaptive network of cytosolic interactions that may be required for injectisome binding. Earlier work from our lab identified two distinct *Ye*SctQ-containing cytosolic protein complexes in living cells (18). However, the full compositions of these and additional SctQ-independent cytosolic complexes are yet to be determined.

Efforts at reconstituting cytosolic injectisome protein complexes *in vitro* have begun to elucidate how these proteins may interact with one another. Native mass spectrometry (MS) experiments showed the ability of *Salmonella enterica* SctQ (*Se*SctQ) to bind *Se*SctL and -N with various stoichiometries. Importantly, however, a $SeSctL_2$-$SeSctN$ heterotrimer was consistently observed in any complex that included these two proteins (19). Dynamic light scattering was used to determine the relative size of cytosolic injectisome proteins and their respective complexes (20). Key differences were observed for reconstituted complexes containing *Shigella flexneri* SctK (*Sf*SctK) and *Sf*SctQ and complexes containing *Sf*SctQ and *Sf*SctL. Specifically, the complex of *Sf*SctK and *Sf*SctQ is globular and compact (hydrodynamic diameter = 10.2 nm), whereas the complex of *Sf*SctQ and *Sf*SctL is much larger (hydrodynamic diameter = 18.5 nm). These differences were attributed to the ability of *Sf*SctQ to adopt different conformational states depending on its binding partner.

To gain further insights into how the cytosolic injectisome protein complexes interact with each other in living *Y. enterocolitica*, we used three-dimensional (3D) single-molecule localization and tracking microscopy to resolve the different diffusive states of *Ye*SctQ, -L, and -N in different genetic backgrounds. We show that deletion of the injectisome protein *Ye*SctD, which presumably binds *Ye*SctK (15), abrogates the clustered, membrane-proximal localization of all cytosolic injectisome proteins and increases the abundance of the cytosolic complexes that are also present in wild-type cells. We further found that deletion of *Ye*SctQ increases the abundance of a complex containing *Ye*SctN and -L and eliminates the diffusive

state assigned to the larger complex containing *Ye*SctN, -L, and -Q. Comparing the diffusion rates of all tracked proteins allowed us to distinguish between oligomerization models that differ in terms of protein complex stoichiometry. Our combined results narrow down the possible stoichiometries of cytosolic injectisome protein complexes that can form prior to injectisome binding.

## RESULTS

**Diffusion of monomeric eYFP under secretion-active conditions.** Tracking of the cytosolic injectisome proteins under secretion-active conditions requires that cells be exposed to a 25°C→37°C temperature jump and that calcium ions be removed from the growth medium through chelation (21–25). Concurrent with the removal of $Ca^{2+}$ ions, $MgCl_2$ and glycerol are added (see Materials and Methods). The altered growth medium composition results in a different osmotic environment for the cells, which could alter the density of the cytosol. To quantify this effect, we tracked monomeric enhanced yellow fluorescent protein (eYFP) under secretion-active conditions in living *Y. enterocolitica* cells. The pooled single-molecule trajectories provide highly sampled distributions of molecular motion behaviors, quantified using the apparent diffusion coefficient ($D^*$). A simulated library of $D^*$ distributions (Fig. 1a) enables linear fitting of experimentally acquired $D^*$ distributions to obtain spectra of intracellular Brownian diffusion coefficients ($D$) that manifest for a given protein. Peaks in these spectra can be analyzed to determine the diffusion rates of different diffusive states, and the areas underneath the peaks determine their relative abundances. These parameters can then be further refined by nonlinear fitting of a defined number of diffusive states (see Materials and Methods). Transforming the raw data (cumulative distribution functions [CDFs] of apparent diffusion coefficients) into diffusion coefficient spectra addresses a key challenge in analyzing single-molecule tracking data, namely, choosing a suitable fitting model among many with different numbers of parameters (diffusive states and their relative abundances) (26, 27).

For monomeric eYFP under secretion-active conditions, we observe a diffusion coefficient spectrum exhibiting a single, well-defined peak at a $D$ value of 9.1 $\mu m^2/s$ that produces a good fit to the experimentally acquired $D^*$ distribution (Fig. 1b and c). Nonlinear fitting using a single diffusive state at a $D$ value of ~9.2 $\mu m^2/s$ results in an equally good fit (Fig. 1b and c). The narrowness of the sole spectral peak shows that the motion of eYFP is homogeneous in the *Y. enterocolitica* cytosol and well described by a single diffusive state with a Brownian diffusion coefficient at ~9.2 $\mu m^2/s$. This property establishes eYFP as a suitable probe for measuring the diffusion coefficients of eYFP-labeled proteins. We note that a diffusive state at a $D$ value of 9.2 $\mu m^2/s$ is slower than the single diffusive state at ~11.9 $\mu m^2/s$, which we measured previously under standard (i.e., nonsecreting) growth conditions (18, 28). This observation is consistent with an overall osmotic upshift that increases the biomolecular density of the cytosol and thus limits the mobility of eYFP. Controlled osmotic upshifts have been previously shown to slow diffusion of fluorescent and fluorescently labeled proteins (29, 30).

Under secretion-active conditions, we also observe a small (7%) population of slow ($D < 0.5$ $\mu m^2/s$) eYFP, indicating that even small proteins can occasionally get trapped, presumably in crowded pockets of the cytosol (29). We verified that trajectories corresponding to slow-moving eYFP are indeed randomly localized to the cytosol and do not show a preference for subcellular locations, such as the membrane or the cell poles (see Fig. S1a in the supplemental material). Notably, the slow population is not detectable when eYFP is tracked under standard growth conditions.

**The diffusive state assignment of *Ye*SctQ indicates two distinct protein complexes in living cells.** Our previously published *Ye*SctQ tracking results for *Y. enterocolitica* under secretion-active conditions suggested the presence of three diffusive states at $D$ values of ~1.0, ~4.0, and ~15 $\mu m^2/s$, with the diffusive state at ~15 $\mu m^2/s$ constituting a 20% population fraction of all tracked proteins. However, such a result is inconsistent with the monomeric eYFP tracking result, as an eYFP-labeled protein should not be able to diffuse faster than 9.2 $\mu m^2/s$ under secretion-active conditions. We therefore reexamined the raw *Ye*SctQ single-molecule trajectories and found that uncharacteristically high apparent diffusion coefficients are often due to large displacements resulting from mislocalization of

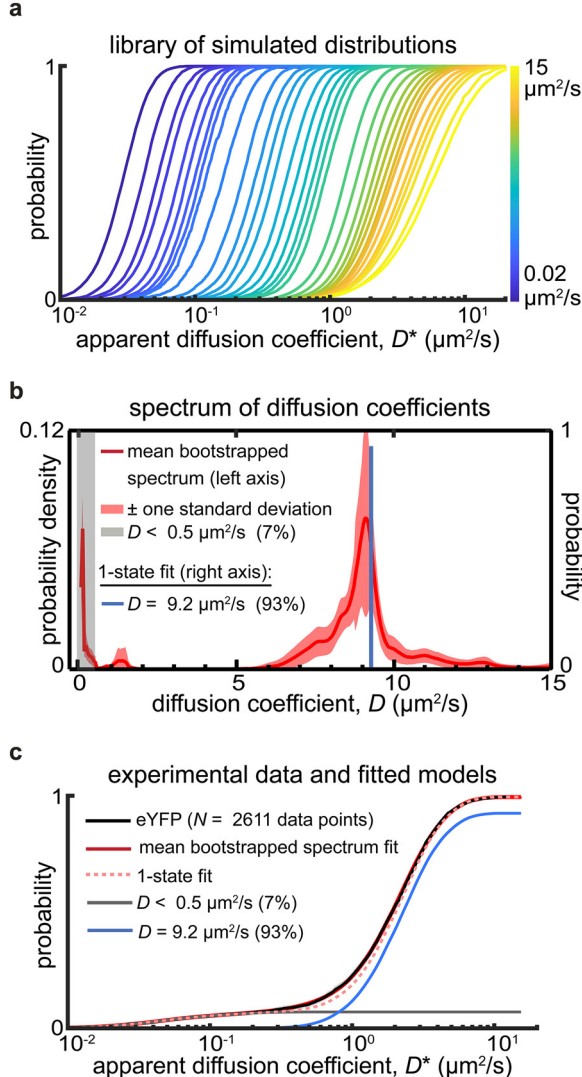

**FIG 1** (a) Simulated distributions of apparent diffusion coefficients based on Monte Carlo simulations of confined Brownian motion in rod-shaped bacterial cells. These simulations account for both random and systematic measurement errors encountered in single-molecule tracking measurements (see Materials and Methods). (b and c) The experimentally measured apparent diffusion coefficient distribution of freely diffusing eYFP molecules is well fitted using a tightly peaked distribution of diffusion coefficients centered at a $D$ value of 9.2 $\mu$m²/s (red curve in panel b) or using a single diffusive state with a $D$ value of 9.2 $\mu$m²/s (blue line in panel c). The excellent agreement between theory and experiment supports that the assumption of bacterial cell-confined Brownian diffusion is valid.

fluorescence signals beyond the axial ($z$) boundaries of the cell (Fig. S2). Removal of these localizations and the associated trajectories removed the fast diffusive state at a $D$ value of ~15 $\mu$m²/s. Importantly, the diffusive states below 9.2 $\mu$m²/s are largely unaffected by this filtering step.

In contrast to freely diffusing eYFP (Fig. 1), the diffusion coefficient spectrum of *Ye*SctQ shows two prominent peaks at $D$ values of 1.3 and 4.8 $\mu$m²/s and a small peak at 6.5 $\mu$m²/s in addition to an injectisome-bound, stationary population at a $D$ value of 0 $\mu$m²/s (Fig. 2a). eYFP-*Ye*SctQ fusion proteins were expressed as a genomic replacement under the control of its native promoter in actively secreting cells, and the fusion protein was stable and not degraded (18). These results are roughly consistent with our previous study (18), which identified three cytosolic diffusive states for *Ye*SctQ at $D$ values of ~1.1, ~4.0, and ~13.9 $\mu$m²/s (the fastest state was assigned to the eYFP-*Ye*SctQ monomer). The distinct peaks in the newly analyzed diffusion coefficient spectrum provide the initial diffusion rates and population fractions for nonlinear fitting of the experimental $D^*$ CDF of *Ye*SctQ. Still, to test whether

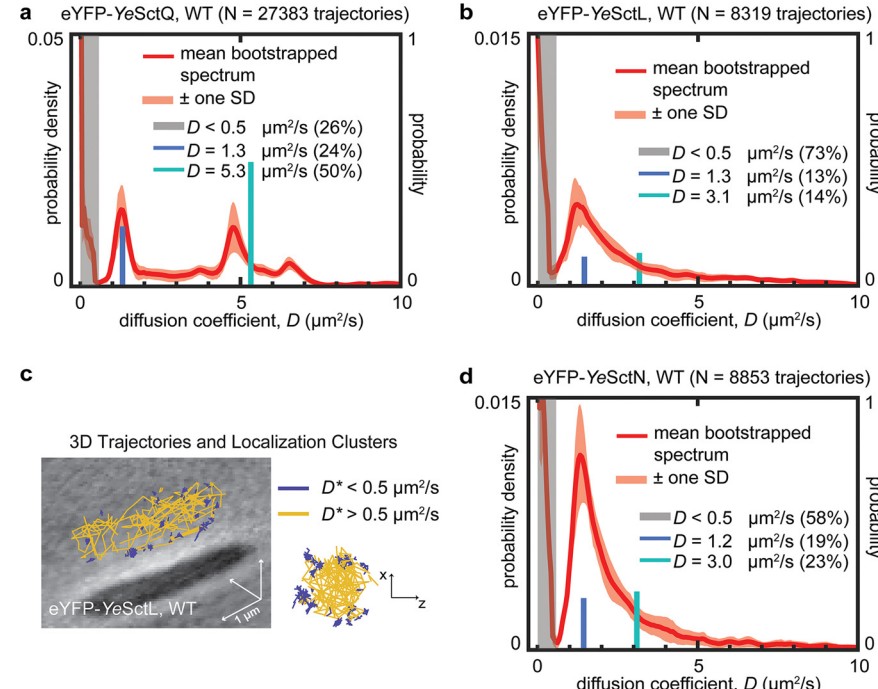

**FIG 2** (a) Diffusion coefficient spectrum for *Ye*SctQ in wild-type cells. The experimentally measured distribution is best fitted with two states at $D$ values of $\sim$1.3 $\mu$m²/s and $\sim$5.3 $\mu$m²/s (Fig. S5). (b) Diffusion coefficient spectrum for *Ye*SctL in wild-type cells. The experimentally measured distribution is best fitted with two states at $D$ values of $\sim$1.3 $\mu$m²/s and $\sim$3.1 $\mu$m²/s (Fig. S6). (c) 3D trajectories for *Ye*SctL with $D^*$ values of $<$0.5 $\mu$m²/s exhibit clustering at the membrane indicative of association with injectisomes. (d) Diffusion coefficient spectrum for *Ye*SctN in wild-type cells. The experimentally measured distribution is best fitted with two diffusive states at $D$ values of $\sim$1.2 $\mu$m²/s and $\sim$3.0 $\mu$m²/s (Fig. S7).

overfitting the data with too many different diffusive states (i.e., a nonlinear model with too many free parameters) could be an issue, we initialized 1-, 2-, and 3-state fits by combining population fractions and initial guesses for diffusion coefficients from the spectrum (see Materials and Methods). Nonlinear fitting using a 2-state model produced diffusive states at $D$ values of $\sim$1.3 and $\sim$5.3 $\mu$m²/s, which agrees with the prominent peaks in the diffusion coefficient spectrum (Fig. 2a), although the $\sim$5.3-$\mu$m²/s diffusive state appears to be right-shifted by the small spectral density at $\sim$6.5 $\mu$m²/s. Nonlinear fitting using a 3-state model produced diffusive states at $D$ values of $\sim$1.2, $\sim$4.4, and $\sim$7.0 $\mu$m²/s; however, the population fraction of the diffusive state at $\sim$7.0 $\mu$m²/s appears to be vastly overestimated (Fig. S5). Additionally, 5-fold cross-validation analysis favors a 2-state over a 3-state model (Fig. S5). These results suggest that the majority of *Ye*SctQ is part of two predominant cytosolic complexes diffusing at $D$ values of $\sim$1.3 and $\sim$4.8 $\mu$m²/s, while the monomeric fraction of this injectisome protein remains small by comparison.

***Ye*SctL shares one diffusive state with *Ye*SctQ.** Having reproduced our previously obtained results in the diffusion coefficient spectra, we next analyzed the intracellular diffusive behaviors of eYFP-labeled *Ye*SctL. We had previously tracked *Ye*SctL with PAmCherry1 but chose to switch the fluorescent label to eYFP for the present study due to its homogeneous diffusion behavior in *Y. enterocolitica* (Fig. 1a and b). Again, we expressed eYFP-*Ye*SctL as a genomic replacement under the control of its native promoter. The eYFP-*Ye*SctL fusion protein was functional and not degraded (Fig. S3 and S4). Examination of the *Ye*SctL diffusion coefficient spectrum reveals a prominent spectral peak at a $D$ value of 1.3 $\mu$m²/s with a small shoulder to the right (Fig. 2b). The 3-state model for *Ye*SctL produced diffusive states at $D$ values of $\sim$1.3 (population fraction = 17%), $\sim$3.2 (population fraction = 3%), and $\sim$7.4 $\mu$m²/s (population fraction = 7%). Given that the diffusion coefficient spectrum clearly indicates a prominent peak at a $D$ value of $\sim$1.3 $\mu$m²/s and does not include a peak at $\sim$0.9 or $\sim$7.4 $\mu$m²/s, we turned to the 2-state model, whose fitted values agree better with the spectral peak positions. The choice of the 2-state model is also supported by 5-fold cross-validation analysis (Fig. S6).

The 2-state model provides two diffusive states for YeSctL at $D$ values of ~1.3 and ~3.1 $\mu$m$^2$/s, with corresponding population fractions of 13% and 14%, respectively (Fig. 2b; Fig. S6). These mobile states are in addition to an abundant (population fraction = 73%) stationary, injectisome-bound population with a $D$ value of <0.5 $\mu$m$^2$/s. Plotting the 3D trajectories using a $D^*$ value threshold of 0.5 $\mu$m$^2$/s clearly shows membrane-associated clusters representing the injectisome-bound population (Fig. 2c). The diffusive state at ~1.3 $\mu$m$^2$/s is consistent with our previous result obtained with PAmCherry-YeSctL, based on which we concluded that both YeSctQ and YeSctL diffuse at the same low rate as part of the same complex (18). The SctLQ interaction is also supported by a number of previous studies (17, 19, 20). On the other hand, the YeSctL diffusive state at a $D$ value of ~3.1 $\mu$m$^2$/s has not been previously observed. The presence of this additional state suggests another homo- or hetero-oligomeric complex containing YeSctL. However, such a complex does not contain YeSctQ, because a diffusive state at a $D$ value of ~3.1 $\mu$m$^2$/s is not observed in the eYFP-YeSctQ data.

**YeSctN and YeSctL share two distinct diffusive states in Y. enterocolitica.** There are numerous reports documenting interactions between SctN and SctL in both flagellar and virulence-associated T3SSs (31–34). SctL binding prevents SctN hexamerization prior to injectisome binding and thereby negatively regulates SctN's ATPase activity, which is the one of the best-established functional roles of the cytosolic injectisome proteins (31, 33, 35–38). To detect the SctNL interaction in living cells, we expressed eYFP-YeSctN as a genomic replacement under the control of its native promoter and acquired YeSctN single-molecule trajectories in actively secreting cells. The eYFP-YeSctN fusion protein was functional and only minimally degraded (Fig. S3 and S4). The resulting diffusion coefficient spectrum shows a very prominent spectral peak at a $D$ value of 1.3 $\mu$m$^2$/s with, similar to the YeSctL diffusion coefficient spectrum, a small shoulder to the right (Fig. 2d). Nonlinear fitting with a 1-state model produces a diffusive state at a $D$ value of ~1.7 $\mu$m$^2$/s. A 2-state model produces diffusive states at $D$ values of ~1.2 and ~3.0 $\mu$m$^2$/s with corresponding population fractions of 19% and 23% (Fig. 2d; Fig. S7), a result that better matches the diffusion coefficient spectrum. The 2-state model is also supported by cross-validation analysis (Fig. S7). The robust emergence of a diffusive state at a $D$ value of ~1.3 $\mu$m$^2$/s, a state that is also observed in YeSctL and YeSctQ measurements (in both the 2- and 3-state models), suggests the formation of a hetero-oligomeric YeSctNLQ complex. The 2-state model produces an intermediate diffusive state at a $D$ value of ~3.0 $\mu$m$^2$/s, and such a diffusive state was also observed for YeSctL ($D$ ~ 3.1 $\mu$m$^2$/s). We thus tentatively assign this diffusive state to a SctNL complex. While the SctNL interaction is supported by a number of previous studies, the population fraction of this state is low for both YeSctN and YeSctL in wild-type cells. We therefore concluded that establishing the existence of this state needed further experimental support.

We argue above that the oligomeric complex that gives rise to the diffusive state at a $D$ value of ~3.0 $\mu$m$^2$/s does not contain YeSctQ. It follows that deletion of sctQ should not affect the formation of that complex. To test this hypothesis, we acquired single-molecule trajectories for YeSctN and YeSctL in a $\Delta$sctQ mutant background. Remarkably, the CDFs of apparent diffusion coefficients for these two proteins (i.e., the raw data) are essentially congruent in the $\Delta$sctQ background, strongly indicating codiffusive behavior (Fig. 3a). Plotting the 3D trajectories for eYFP-YeSctN in the $\Delta$sctQ background using a $D^*$ threshold of 0.5 $\mu$m$^2$/s shows an absence of membrane-associated clusters (Fig. 3b). The same phenomenon was observed for eYFP-YeSctL in the $\Delta$sctQ background (data not shown), indicating that injectisome association is no longer possible in the absence of SctQ (17). The spectra of diffusion coefficients of these two proteins show the same trend: each spectrum contains a prominent peak at a $D$ value of 2.8 $\mu$m$^2$/s. Notably, a peak at 1.3 $\mu$m$^2$/s, which we previously assigned to a YeSctNLQ complex, is also absent. A 1-state nonlinear fit converged on a $D$ value of ~2.7 $\mu$m$^2$/s in each case, with corresponding population fractions of 93% for YeSctL and 92% for YeSctN (Fig. S8 and S9).

Together, these results support the conclusion that YeSctL and YeSctN are able to form two distinct cytosolic complexes. One of these complexes contains YeSctQ and the other does not. Under our experimental conditions, the YeSctNLQ complex diffuses at a $D$ value

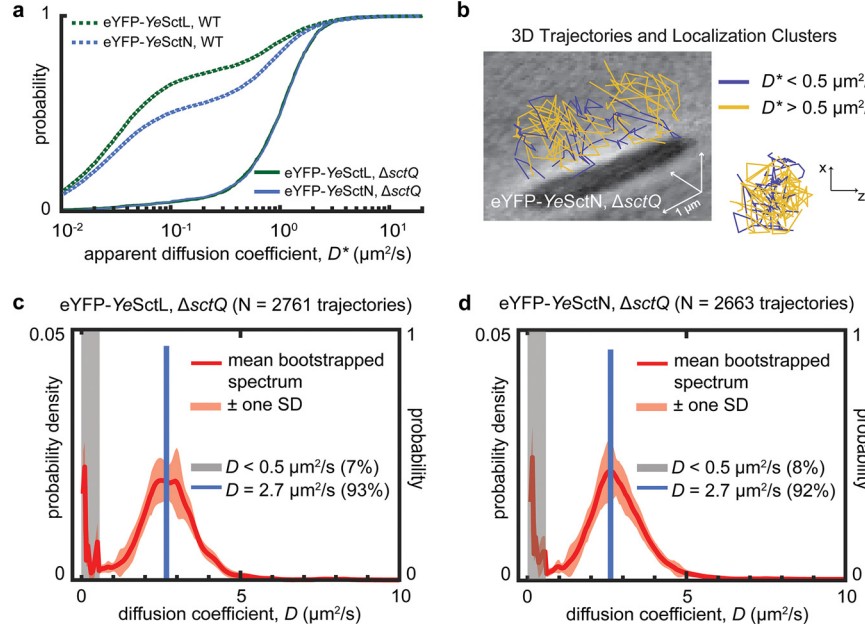

**FIG 3** (a) Experimentally measured CDFs of *Ye*SctL and *Ye*SctN apparent diffusion coefficients in wild-type cells and Δ*sctQ* mutants. The *Ye*SctL and *Ye*SctN distributions overlay very well in the Δ*sctQ* background. (b) 3D trajectories for *Ye*SctN in the Δ*sctQ* background show an absence of membrane-associated clusters for $D^*$ values of <0.5 $\mu$m²/s. (c and d) Diffusion coefficient spectra for *Ye*SctL and *Ye*SctN in the Δ*sctQ* background. In each case, the experimentally measured distribution is best fitted with one state at a $D$ value of ~2.7 $\mu$m²/s (Fig. S8 and S9).

of ~1.3 $\mu$m²/s, while the *Ye*SctNL complex diffuses at ~2.8 $\mu$m²/s. In wild-type cells, the presence of *Ye*SctQ leads to a smaller abundance of the *Ye*SctNL complex in favor of the *Ye*SctNLQ complex as well as injectisome-bound *Ye*SctL and *Ye*SctN.

**Native-complex formation among *Ye*SctQ, -L, and -N increases in the absence of injectisomes.** SctD is the inner-membrane ring protein of the injectisome and is required for proper injectisome assembly (39) and subsequent effector protein secretion. SctD provides a binding interface for SctK, which in turn allows SctQ, -L, and -N to bind to the injectisome (40). Thus, SctQ, -L, and -N are completely cytosolic in the Δ*sctD* background. To test whether the same diffusive states are manifested in the absence of fully assembled injectisomes, we acquired single-molecule trajectories in the Δ*sctD* mutant background for *Ye*SctQ, -L, and -N.

The *Ye*SctQ Δ*sctD* spectrum is very similar to the spectrum observed in wild-type cells. Two clear spectral peaks are evident at $D$ values of 1.3 and ~4.5 $\mu$m²/s (Fig. 4a; Fig. S10). A 2-state nonlinear fit produces diffusive states at $D$ values of ~1.2 and ~4.5 $\mu$m²/s, a result that is in excellent agreement with the diffusive state observed in wild-type cells. These data show that the same complexes formed in wild-type cells can also form in the absence of injectisomes. Consistent with the loss of injectisome binding, the combined population fractions of the three distinct cytosolic diffusive states increase. The state(s) at a $D$ value of <0.5 $\mu$m²/s decreased from 24% in wild-type cells to 10% in the Δ*sctD* mutant. We note, however, that the 10% slowly diffusing molecules in the Δ*sctD* mutant do not localize to the membrane (Fig. S1b).

The *Ye*SctL and *Ye*SctN Δ*sctD* spectra both show a broad peak extending from a $D$ value of ~1 to ~4 $\mu$m²/s. A 2-state fit converges to diffusive states at $D$ values of ~1.2 and ~2.6 $\mu$m²/s for *Ye*SctL and ~1.0 and ~2.5 $\mu$m²/s for *Ye*SctN. Both of these results agree well with the diffusion coefficient spectra (Fig. 4b and c; Fig. S11 and S12) and are also in close agreement with the diffusive states observed for each protein in wild-type and Δ*sctQ* cells. Taken together, these results establish that the same complexes formed in wild-type cells can also form, in increased abundance, in the absence of injectisomes. In other words, the presence of injectisomes is not necessary for cytosolic complex formation.

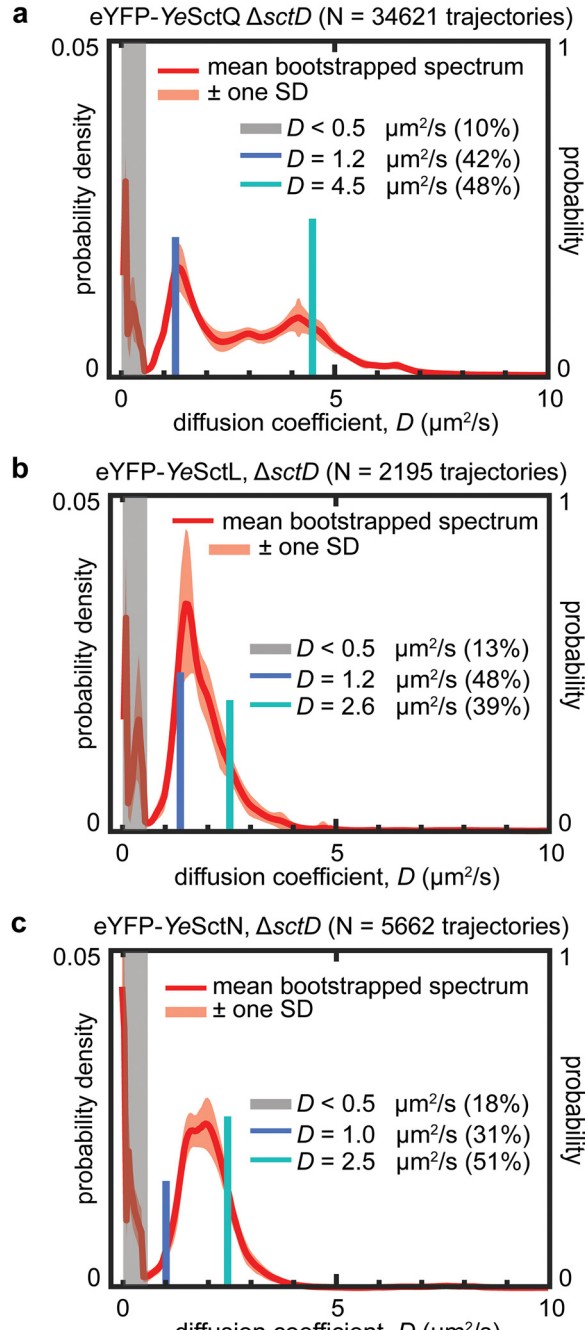

**FIG 4** Removal of *Ye*SctD results in an increased abundance of native cytosolic complexes. *Ye*SctQ (a), *Ye*SctL (b), and *Ye*SctN (c) all share a diffusive state around a *D* value of ~1.3 $\mu$m²/s, consistent with results obtained in wild-type cells. Also consistent with wild-type results is the presence of a diffusive state at a *D* value of ~4.5 $\mu$m²/s for *Ye*SctQ and the presence of a shared diffusive state for *Ye*SctL and *Ye*SctN at ~2.5 $\mu$m²/s.

## DISCUSSION

Determining the functional role(s) of the cytosolic injectisome proteins has been challenging, because our understanding of their interactions at the injectisome and in the cytosol remains limited. Here, we present live-cell single-molecule tracking data that suggest the formation of distinct, freely diffusing protein complexes. One complex containing *Ye*SctN and *Ye*SctL, but not *Ye*SctQ, diffuses at a *D* value of ~2.8 $\mu$m²/s. A larger complex containing *Ye*SctN, -L, and -Q diffuses at a *D* value of ~1.3 $\mu$m²/s. These complexes form robustly in wild-type and mutant cells that are unable to fully assemble T3SS injectisomes (Fig. 5a).

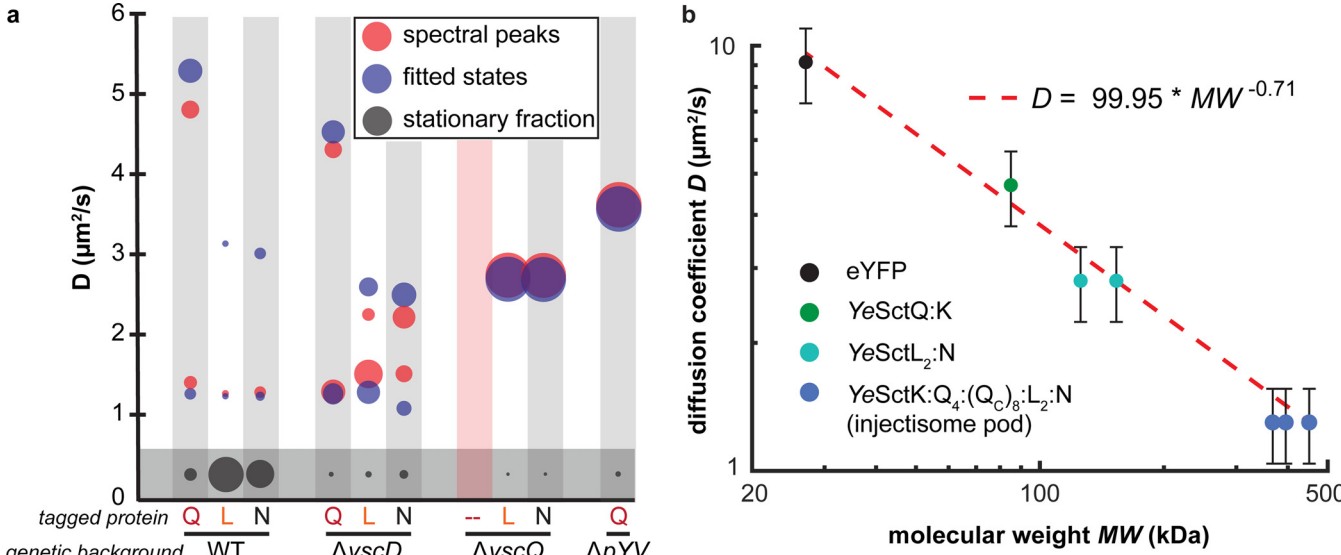

**FIG 5** (a) Comparison of spectral peaks and fitted diffusive states for each indicated eYFP-labeled injectisome protein tracked in different genetic backgrounds. The relative population fraction for each peak/diffusive state is represented by the size of the circle. Spectra for eYFP-*Ye*SctN and eYFP-*Ye*SctL in the Δ*sctD* mutant do not contain prominent spectral peaks corresponding to the fitted diffusive state at a *D* value of ~2.8 $\mu$m²/s, whereas prominent spectral peaks are resolved in the Δ*sctQ* mutant. The fitted diffusive states for a *D* value of <0.5 $\mu$m²/s in the Δ*sctD*, Δ*sctQ*, and Δ*pYV* mutants do not localize to the membrane. (b) Measured diffusion coefficients plotted against the molecular weights of the indicated protein complexes. The straight line corresponds to the least-squares fit using the model shown.

A *Ye*SctNL complex is consistent with a proposed regulatory mechanism of *Ye*SctL. SctL is thought to negatively regulate ATPase activity in both bacterial flagellar motors and injectisomes by preventing SctN hexamerization in the cytosol (33). Recent cryo-EM data on purified proteins show that the extreme N terminus of FliI (the flagellar homologue of SctN) contains a charged region to which FliH (the flagellar homologue of SctL) can bind (41). Abolishing this interaction through point mutations results in a higher propensity of ATPase hexamerization *in vitro*, as well as decreased cell growth and cell motility phenotypes. Further, the SctNL (FliIH) complex has been consistently observed as a heterotrimer using *in vitro* bio-chemical and biophysical approaches (11, 19, 31, 36, 37, 42, 43), with a dimer of SctL (FliH) binding to the ATPase monomer SctN (FliI). Based on these results, we posit that the diffusive state at a *D* value of ~2.8 $\mu$m²/s that we observed in both wild-type cells and Δ*sctQ* mutants and the diffusive state at ~2.5 $\mu$m²/s that we observed in Δ*sctD* mutants is likely due to a *Ye*SctN$_1$-*Ye*SctL$_2$ heterotrimer. In wild-type cells, 19% of *Ye*SctN and 13% of *Ye*SctL contribute to this heterotrimer population. In Δ*sctQ* and Δ*sctD* mutants, the abundance of this hetero-trimer increases substantially (93% and 92% for *Ye*SctN and *Ye*SctL, respectively, in the Δ*sctQ* mutant and 31% and 48% for *Ye*SctN and *Ye*SctL, respectively, in the Δ*sctD* mutant).

A *Ye*SctNLQ complex is consistent with cytosolic pod complexes containing SctN, -L, -Q, and -K. Subtomogram averages in *Shigella* and *Salmonella* suggest six distinct pods composed of SctK, -Q, and -L that seem to cradle the hexameric ATPase at the injectisome (9, 10). Subsequent work has focused on reconstituting such pod complexes *in vitro*. *Se*SctNLQ (19) and *Sf*SctLQK (20) complexes were successfully isolated, and the estimated size and shapes of these complexes were qualitatively consistent with cryo-ET pod densities. Based on these results, we speculate that the diffusive state at a *D* value of ~1.3 $\mu$m²/s that we observed in wild-type cells and in Δ*sctD* mutants is likely not just a SctNLQ complex but in fact a SctNLQK heterooligomer. This claim is substantiated by the loss of the diffusive state at a *D* value of ~1.3 $\mu$m²/s when we tracked eYFP-*Ye*SctL and eYFP-*Ye*SctN proteins in the Δ*sctQ* mutant. Whether a diffusive state at a *D* value of ~1.3 $\mu$m²/s is also manifested for SctK in wild-type cells and in Δ*sctD* mutants remains to be determined.

The molecular composition of the diffusive state at a *D* value of ~4.8 $\mu$m²/s observed for eYFP-*Ye*SctQ could be consistent with a *Ye*SctQK complex. In our previous work (18), we assigned this intermediate diffusive state to a *Ye*SctQ homo-oligomer that also con-tains *Ye*SctQ$_C$, the alternatively expressed C-terminal fragment of *Ye*SctQ. This assignment

was made based on tracking eYFP-*Ye*SctQ proteins in a Δ*pYV* mutant, which does not express any other T3SS proteins. In that mutant, ~80% of the tracked *Ye*SctQ proteins diffused at a $D$ value of ~3.6 $\mu$m²/s, which is notably smaller than the diffusive state at a $D$ value of ~4.8 $\mu$m²/s that we observe in wild-type cells. This discrepancy indicates that the diffusive states at $D$ values of ~3.6 and 4.8 $\mu$m²/s likely originate from different complexes. The spectrum of eYFP-*Ye*SctQ diffusion coefficients in a Δ*pYV* background indeed show a prominent spectral peak ranging from a $D$ value of ~3 to ~4.5 $\mu$m²/s but little spectral density at ~4.8 $\mu$m²/s (Fig. S12). *In vitro* light-scattering measurements on *Sf*SctQ-containing complexes (20) suggested the ability of *Sf*SctQ to adopt different conformational states depending on its interacting partner. A small, globular complex (hydrodynamic diameter = 10.2 nm) was observed upon coexpression and copurification of *Sf*SctQ and -K, while a larger complex (hydrodynamic diameter = 13.4 nm) was observed for copurified *Sf*SctQ and -$Q_C$. Based on these results, we speculate that the diffusive state at a $D$ value of ~4.8 $\mu$m²/s in wild-type cells and in Δ*sctD* mutants is due to a *Ye*SctQK complex. In wild-type cells, we observe very little spectral density in the range of $D$ values from 3.5 to 4 $\mu$m²/s, which indicates that a *Ye*SctQQ$_C$ complex is not abundantly present. On the other hand, if injectisomes are not assembled, as in the Δ*sctD* mutants, we do observe spectral density in the $D$-value range from 3.5 to 4 $\mu$m²/s. These results indicate that the presence of injectisomes and thus the ability of *Ye*SctQ to bind to and unbind from injectisomes affects the type of complexes that *Ye*SctQ forms in abundance. Determining the compositions and stoichiometries of *Ye*SctQ-containing complexes in specific deletion mutants will be the subject of future work.

Complex formation among cytosolic injectisome proteins may provide the foundation for functional regulation of type 3 secretion. Specifically, a pool of freely diffusing complexes of injectisome proteins that are available for injectisome binding would provide a mechanism for timely reactivation of secretion. Recent work has shown that an extracellular pH drop from 7 to 4 results in disassociation of *Ye*SctN, -L, -Q, and -K from injectisomes, an effect that correlated with loss of secretion (44). Notably, the intracellular pH decreased only slightly (pH = 7→6.3) in acidic extracellular environments. Reverting the external pH from 4 to 7 reverses these effects. Injectisome binding is restored within ~10 min, and effector protein secretion resumes shortly thereafter. The results reported here lend further support for a secretion reactivation model relying on cytosolic complexes of injectisome proteins: in Δ*sctQ* and in Δ*sctD* mutants, in which injectisome binding is not possible, we observe clear signatures of the same diffusive states in wild-type cells that we assign to distinct *Ye*SctNLQ, *Ye*SctNL, and *Ye*SctQK complexes.

Identifying which cytosolic injectisome proteins assemble into freely diffusing complexes in a native biological context can help determine how these proteins bind and unbind dynamically to and from injectisomes. While we have identified the compositions of three prominent complexes, the exact stoichiometries and functions of these complexes remain to be determined. Previous work by others has shown that the cytosolic diffusion rate of proteins scales with molecular weight but not according to the Stokes-Einstein equation, which stipulates the following: $D \propto$ (molecular weight)$^{-1/3}$. Instead, a scaling expressed as $D \propto$ (molecular weight)$^{-2/3}$ has been observed by others independently for different proteins (45–48). Here, we obtained a scaling expressed as $D \propto$ (molecular weight)$^{-0.71}$ when plotting the measured diffusion coefficients against the molecular weights of monomeric eYFP and the protein complexes discussed above (likely stoichiometries were estimated based on current biophysical models of the cytosolic injectisome protein complexes) (Fig. 5b) (20, 34). This analysis allows us to distinguish between oligomerization models that differ in terms of protein complex stoichiometry. For example, earlier studies suggested that SctK, -Q, -$Q_C$, -L, and possibly -N form a high-molecular-weight (>1-MDa) complex, termed the sorting platform, that sequentially interacts with different secretion substrates and their chaperones (49, 50). However, efforts aimed at reconstituting sorting platform complexes *in vitro* have not been reported. The analysis in Fig. 5b suggests that the majority of cytosolic injectisome proteins exist as part of smaller complexes with molecular weights of less than 500 kDa. Our combined results thus narrow down the possible stoichiometries of cytosolic injectisome protein complexes that can form prior to injectisome binding.

Future work will need to determine the molecular structures of the complexes identified here and how their individual abundances change in response to environmental signals that induce T3SS assembly and activate secretion. However, as shown in this study, careful attention must be paid to the effect of the osmotic environment on cytosolic protein diffusion rates. Such efforts will help determine how the cytosolic injectisome proteins interact in the cytosol and when bound to injectisomes and how their dynamic exchange at injectisomes contributes to functional regulation of secretion.

## MATERIALS AND METHODS

**Bacterial strains.** *Yersinia enterocolitica* strains expressing fluorescent fusion proteins were generated by allelic exchange as previously described (39, 51). Mutator plasmids containing 250- to 500-bp flanking regions, the coding sequence for eYFP, and a 13-amino-acid flexible linker region between the fluorescent and target protein were introduced into *E. coli* SM10 $\lambda$pir for conjugation with *Y. enterocolitica* pIML421asd. After sucrose counterselection for the second allelic exchange event, colonies were grown overnight in brain heart infusion (BHI) medium (Sigma-Aldrich, St. Louis, MO) with nalidixic acid (Sigma-Aldrich) (35 $\mu$g/mL) and 2,6-diaminopimelic acid (Chem Impex International, Wood Dale, IL) (80 $\mu$g/mL). PCR screening was then performed to confirm target insertion, and constructs were confirmed by sequencing (Genewiz, South Plainfield, NJ).

**Cell culture.** *Yersinia enterocolitica* strains were inoculated from freezer stock 1 day prior to imaging and grown overnight in BHI medium containing nalidixic acid (Nal) (35 $\mu$g/mL) and 2,6-diaminopimelic acid (DAP) (80 $\mu$g/mL) at 28°C with shaking. On the day of imaging, 250 $\mu$L of overnight culture was transferred into fresh BHI medium containing Nal and DAP and grown at 28°C with shaking for 1 h. Glycerol (4 mg/mL), $MgCl_2$ (20 mM), and EDTA (5 mM) were then added to the culture medium, and the culture was transferred to a 37°C water bath with shaking for 3 h to induce expression of the *yop* regulon and ensure secretion activation. Cells were pelleted by centrifugation at 5,000 $\times$ *g* for 3 min and resuspended three times in fresh M2G (4.9 mM $Na_2HPO_4$, 3.1 mM $KH_2PO_4$, 7.5 mM $NH_4Cl$, 0.5 mM $MgSO_4$, 10 $\mu$M $FeSO_4$ [EDTA chelate; Sigma], 0.5 mM $CaCl_2$) with 0.2% glucose as the sole carbon source. After the three washes, the remaining pellet was resuspended in M2G, DAP, $MgCl_2$, glycerol, and EDTA. Cells were finally plated on 1.5 to 2% agarose pads in M2G containing DAP, glycerol, and $MgCl_2$.

**Secretion assay and protein analysis.** Cultures were inoculated to an optical density at 600 nm ($OD_{600}$) of 0.12 from stationary overnight cultures. After 1.5 h of growth at 28°C, induction of the *yop* regulon was performed by shifting the culture to 37°C. Cultures were further incubated at 37°C for 3 h. Bacteria from 2 mL culture were collected (15,000 $\times$ *g*, 10 min, 4°C). Then, 1.8 mL of supernatant was mixed with 0.2 mL trichloroacetic acid (final concentration, 10%) and incubated overnight at 4°C to precipitate proteins within the supernatant. Proteins were collected (15,000 $\times$ *g*, 15 min, 4°C) and washed twice with ice-cold acetone (15,000 $\times$ *g*, 5 min, 4°C). The pellet was dried at room temperature for 1 h. The pellet was resuspended in SDS-PAGE loading buffer (SDS [2%, wt/vol], Tris-HCl [0.1 M], glycerol [10%, wt/vol], dithiothreitol [DTT; 0.05 M]; pH 6.8) and 0.6 OD units (1 OD unit is the equivalent of 1 mL culture at an $OD_{600}$ of 1) in 15 $\mu$L were used for the SDS-PAGE gel analysis. For Western blot analysis of the total cellular sample, the collected bacterial pellet was prepared for Western blot analysis by directly normalizing to 0.3 OD units in 15 $\mu$L loading buffer. All samples were heated for 10 min at 99°C before loading. Proteins were separated by SDS-PAGE on 15% acrylamide gels. For visualization, the gels were stained with InstantBlue (Expedeon). For immunoblots, the proteins were blotted on a nitrocellulose membrane. Detection of the eYFP tag was performed by using primary rabbit antibodies against green fluorescent protein (GFP; 1:5,000) (Invitrogen A6455, lot number 1853896) and secondary anti-rabbit immunoglobulin antibodies conjugated with horse-radish peroxidase (1:10,000) (Sigma A8275). The immunoblot was visualized using enhanced chemiluminescence (ECL) substrate (Pierce) on a LAS-4000 luminescence image analyzer (Fujifilm).

**Single-molecule superresolution fluorescence imaging.** Image data were acquired on a custom-built inverted fluorescence microscope based on the RM21 platform (Mad City Labs, Inc, Madison, WI), as previously described (18, 28). Immersion oil was placed between the lens objective (UPLSAPO 100; numerical aperture [NA], 1.4) and the glass coverslip (VWR, Radnor, PA; number 1.5; 22 mm by 22 mm). Single-molecule images were obtained by utilizing eYFP photoblinking (52). A 514-nm laser (Coherent, Santa Clara, CA; Genesis MX514 MTM) was used for excitation of eYFP ($\sim$350 W cm$^{-2}$). Zero-order quarter-wave plates (Thorlabs, Newton, NJ; WPQ05M-405, WPQ05M-514, and WPQ05M-561) were used to circularly polarize all excitation lasers, and the spectral profile of the 514-nm laser was filtered using a bandpass filter (Chroma, Bellows Falls, VT; ET510/10bp). Fluorescence emission from both eYFPs was passed through a shared filter set (Semrock, Rochester, NY; LP02-514RU-25, Semrock NF03-561E-25, and Chroma ET700SP-2P8). The emission path contains a wavelength-specific dielectric phase mask (Double Helix, LLC, Boulder, CO) that is placed in the Fourier plane of the microscope to generate a double-helix point spread function (DHPSF) (53, 54). The fluorescence signal is detected on a scientific complementary metal oxide semiconductor (sCMOS) camera (Hamamatsu, Bridgewater, NJ; ORCA-Flash 4.0 V2). Up to 20,000 frames are collected per field of view with an exposure time of 25 ms. A flip mirror in the emission pathway enables toggling the microscope between fluorescence imaging and phase-contrast imaging modes without having to change the lens objective of the microscope.

**Data processing.** All fluorescence images were processed in MATLAB using a modified version of the easyDHPSF code (18, 28, 55). To extract 3D localizations, fluorescence intensity from single-molecule emitters was fitted to a double-Gaussian PSF model with maximum-likelihood estimation. A median filter with a time window of 10 frames was used for background subtraction.

For each field of view, cell outlines were generated based on the phase-contrast images using the open-source software OUFTI (56). Single-molecule localizations were then overlaid and aligned with the cell outlines.

To ensure that single-molecule localizations align well with the corresponding cell, the cell outlines were registered to the fluorescence data by a two-step 2D affine transformation using the cp2tform function in MATLAB. Five control points were manually selected based on the position of the cell poles of single-molecule localization data, which generated an initial transformation that allowed the removal of any cell containing fewer than 10 localizations. The center of mass for all the remining cells were then used to create a second, larger set of control pairs to compute the final transformation function. Only localizations within cell outlines were considered for further analysis.

**Single-molecule tracking analysis.** 3D single-molecule localizations were filtered (Fig. S2) and linked into trajectories with a distance threshold of less than 2.5 $\mu$m between subsequent localizations. If two or more localizations were present in the cell at the same time, the trajectory was not considered for further analysis to prevent the incorporation of two or more molecules into the same trajectory. Trajectories containing fewer than 4 localizations were also not considered for further analysis.

Trajectory information was then used to calculate the mean square displacement (MSD), as follows:

$$\mathrm{MSD} = \frac{1}{N}\sum_{n=2}^{N}(x_n - x_{n-1})^2$$

where $N$ is the total number of time points and $x_n$ is the 3D position of the molecule at time point $n$. The apparent diffusion coefficient, $D^*$, of a given molecule trajectory was then computed as MSD/($2 \times m \times \Delta t$), where $m$ is the dimensionality of the data and is equal to 3, and $\Delta t$ is the camera exposure time and was 25 ms for all experiments reported here.

**Monte Carlo simulations.** To resolve the unconfined diffusion coefficients of distinct molecular complexes in living cells based on the experimentally measured distribution of apparent diffusion coefficients, we simulated confined Brownian motion trajectories inside a cylindrical volume (radius = 0.4 $\mu$m, length = 5 $\mu$m). We added to our existing pool (28) of simulated diffusion coefficients to improve data fitting (see below). Noisy, motion-blurred single-molecule images mimicking the raw experimental data were simulated for confined (in the cylindrical volume) single molecules with defined Brownian diffusion coefficients. These images were then processed and linked into trajectories in the same manner as our experimental data. The resulting simulated CDFs account for confinement effects of the bacterial cell volume, signal integration over the camera exposure time, and experimentally calibrated signal-to-noise levels. Analyzing the simulation data in the same manner as experimental data ensures that static and dynamic localization errors (28) are accurately modeled for our data fitting routine.

**Data fitting.** Experimental distributions of apparent diffusion coefficients were fitted using a linear combination of simulated CDFs, where each CDF corresponds to a single diffusive state described by a single diffusion coefficient. The coefficients of the best-fitting linear combination were determined using the MATLAB function lsqlin() with the trust-region-reflective algorithm (MathWorks, Inc., Natick, MA). The resulting linear coefficients can be displayed as a spectrum of diffusion coefficients that are manifested for a tracked protein in living cells. To establish the robustness of individual peaks in the diffusion coefficient spectrum, we resampled the raw data using bootstrapping ($N$ = 100). We then averaged these bootstrapped spectra and used the resulting linear coefficients to fit the experimental data (mean bootstrapped spectrum fit). The mean bootstrapped spectrum provides us with an initial estimate of the diffusion coefficients of prominent diffusive states and their relative population fractions, estimated by determining the peak maxima and the area under the spectral peaks, respectively. The parameters thus obtained are then used as input parameters for nonlinear fitting using the particleswarm() function in MATLAB. This approach allowed us to estimate the diffusion coefficients and population fractions of distinct diffusive states that are manifested in living cells. In the cases where the spectrum contained overlapping peaks or featureless nonzero spectral density, we chose one or multiple diffusive states to initialize the nonlinear fitting process. This approach resulted in the $n$-state ($n$ = 1 to 3) fitting models shown in Fig. S3 to S10, which were then evaluated using 5-fold cross-validation and qualitative agreement with the diffusion coefficient spectrum obtained by linear fitting.

**Data availability.** Data is made freely-available using the Open Science Framework (https://osf.io/8hbk4/).

## SUPPLEMENTAL MATERIAL

Supplemental material is available online only.
**SUPPLEMENTAL FILE 1**, PDF file, 2.4 MB.

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
