## [Reviewer comments · Microbiology Spectrum]

Microbiology Spectrum

Distinct cytosolic complexes containing the type III secretion system ATPase resolved by 3D single-molecule tracking in live *Yersinia enterocolitica*

Joshua Prindle, Yibo Wang, Julian Rocha, Andreas Diepold, and Andreas Gahlmann

Corresponding Author(s): Andreas Gahlmann, University of Virginia

Review Timeline:

Submission Date:	May 25, 2022
Editorial Decision:	July 24, 2022
Revision Received:	October 18, 2022
Accepted:	October 20, 2022

Editor: Giordano Rampioni

Reviewer(s): The reviewers have opted to remain anonymous.

Transaction Report:

DOI: <https://doi.org/10.1128/spectrum.01744-22>

July 24, 2022

Prof. Andreas Gahlmann
University of Virginia
409 McCormick Rd.
Room 212
Charlottesville, Virginia 22904

Re: Spectrum01744-22 (Distinct complexes containing the cytosolic type III secretion system ATPase resolved by 3D single-molecule tracking in live *Yersinia enterocolitica*)

Dear Prof. Andreas Gahlmann:

Thank you for submitting your manuscript to Microbiology Spectrum. I am extremely sorry for the late response. This is mainly due to the unavailability of many experts in the field (more than 15!) to act as reviewers of your manuscript. Your manuscript has been finally evaluated by two reviewers with expertise in the area addressed in your study and it was the consensus view of these reviewers that your paper contains interesting data with significant potential impact. However, both reviewers recommend modifications before manuscript acceptance. I will be glad to consider for publication in Microbiology Spectrum a revised version of your manuscript addressing the comments raised by the reviewers.

Link Not Available

Sincerely,

Giordano Rampioni

Journals Department
Reviewer comments:

Reviewer #1 (Comments for the Author):

In this manuscript, Prindle and colleagues studied the complexes formed by cytosolic components of the type III secretion system in living *Yersinia*. To access the nature, dynamism and requirements of these complexes, they characterized the diffusive states of these proteins, a method they previously established. This work is incremental and now brings new insight

into the hetero-oligomeric states of T3SS components prior to interaction with the membrane apparatus. Notably the authors identified a large cytosolic complex including the SctN ATPase, SctL and SctQ that forms independently of the T3SS membrane part.

The manuscript is well written and concise. The single-molecule super-resolution fluorescence imaging approach is sound. I only have two main comments.

First, in order to express eYFP fusions with SctL, N and Q as close as possible to native protein levels, the authors engineered chromosomal fusions at the native locus. However, this is no guarantee of success. If for any reason, the stability or production of the fusions is way off compared to native proteins, this could largely affect the populations and dynamic behaviors observed. I would therefore recommend a more careful analysis of fusion protein levels compared to wild-type.

Second, based on the results, the authors proposed that secretion may be regulated by the assembly of cytosolic complexes prior to incorporation into the membrane complex. I really wish the authors went a little bit further to substantiate that compelling idea, unless the experiment required is too complicated to setup. If the hypothesis is correct and reactivation of secretion relies on the formation of cytosolic complexes in permissive conditions, it would be interesting to test if the dissociation of N, L, Q and K from injectisomes, (just after a pH drop as suggested by the authors), releases free or lower order complexes rather than assembled NLQK complexes which could be readily re-incorporated into injectisomes.

Minor comments in no particular order:

The way the fusions are named, ie eYFP-YeSctQ, makes it complicated to read, and really, is not useful here.

Starting L133 and then L147

Why don't the authors measure the diffusion rate of eYFP in the same conditions as with eYFP-YeSctQ instead of assuming that the diffusion rate of free monomeric eYFP-YeSctQ is faster than that of eYFP due to the different experimental conditions?

L170

Supplementary Figure 2: the legend of the figures, particularly Sup Fig. 2, is not clear and one has to guess what the authors are showing precisely. Molecular mass (numbers) are not indicated in Sup Fig 1. eYFP theoretical mass is 27 kDa and unfortunately, a band is present around that on the gel. In addition, for all three fusions, there are additional bands compared to the WT control (if I understand correctly...). Therefore, I don't see how the authors can rule out the possibility that the peak corresponding to fast diffusing molecules is in fact due to partial cleavage of the fusion.

L268

are (in) also in close agreement

L309 IN both wild-type and Δ sctQ mutants

L269

Would it be clearer to state that the presence of injectisomes is not necessary to form the cytosolic complexes observed? OR that LN and LNQ complexes form prior to incorporation into injectisomes?

L321

I don't understand why this argues in favor of a NLQK complex? Doesn't the absence of Q disrupt the NLQ complex, irrespective of the presence of K?

Reviewer #2 (Comments for the Author):

The article provides information and proof about the composition and the dynamics of the *Yersinia enterocolitica* sorting platform, including its ATPase SctN. The data indicate the formation of diffusing, cytosolic sorting platform complexes prior to T3SS binding. Overall the paper is sound and it is easy to follow the conclusions the authors drew from the obtained results. Anyhow, it would be important to address the major control experiment mentioned below. In addition, addressing and discussing the minor points would make the manuscript even more comprehensive and could open up discussing points that might be addressed in the future.

Major point:

I suggest to perform an additional control experiment. As already mentioned in lines 147-150, the fitted diffusion rate of 14.8 $\mu\text{m}^2/\text{s}$ is faster than the diffusion rate of free eYFP. An additional control experiment under secretion conditions should give further insight if the faster diffusion is really only due to the different conditions. If this is not the case (faster diffusion than free eYFP also under same conditions), please discuss this or potentially use another fitting.

Minor points:

Lines 225-228 and 262-264: Please discuss the reason why there are still slow diffusing molecules ($D < 0.5 \mu\text{m}^2/\text{s}$) in the ΔSctQ and ΔSctD background. In prior experiments this diffusion coefficient was described as stationary/membrane-associated. Here, membrane-associated clusters are absent, but there are still trajectories with $D < 0.5 \mu\text{m}^2/\text{s}$. This is also visible in Fig. 2b compared to 3b and Suppl. Fig. 11.

Please discuss in more detail the differences between the present study and Rocha et al., 2018. Clarify the importance of the inclusion of SctN in this study.

Staff Comments:

Preparing Revision Guidelines

Please return the manuscript within 60 days; if you cannot complete the modification within this time period, please contact me. If you do not wish to modify the manuscript and prefer to submit it to another journal, please notify me of your decision immediately so that the manuscript may be formally withdrawn from consideration by Microbiology Spectrum.

The article provides information and proof about the composition and the dynamics of the *Yersinia enterocolitica* sorting platform, including its ATPase SctN. The data indicate the formation of diffusing, cytosolic sorting platform complexes prior to T3SS binding. Overall the paper is sound and it is easy to follow the conclusions the authors drew from the obtained results. Anyhow, it would be important to address the major control experiment mentioned below. In addition, addressing and discussing the minor points would make the manuscript even more comprehensive and could open up discussing points that might be addressed in the future.

Major point:

I suggest to perform an additional control experiment. As already mentioned in lines 147-150, the fitted diffusion rate of $14.8 \mu\text{m}^2/\text{s}$ is faster than the diffusion rate of free eYFP. An additional control experiment under secretion conditions should give further insight if the faster diffusion is really only due to the different conditions. If this is not the case (faster diffusion than free eYFP also under same conditions), please discuss this or potentially use another fitting.

Minor points:

Lines 225-228 and 262-264: Please discuss the reason why there are still slow diffusing molecules ($D < 0.5 \mu\text{m}^2/\text{s}$) in the ΔSctQ and ΔSctD background. In prior experiments this diffusion coefficient was described as stationary/membrane-associated. Here, membrane-associated clusters are absent, but there are still trajectories with $D < 0.5 \mu\text{m}^2/\text{s}$. This is also visible in Fig. 2b compared to 3b and Suppl. Fig. 11.

Please discuss in more detail the differences between the present study and Rocha et al., 2018. Clarify the importance of the inclusion of SctN in this study.

We thank all the Reviewers for their thoughtful comments (listed below in black font) and the editors for allotting us the time needed to perform the requested control experiments. Our responses to the Reviewer comments are in blue font. Sections from the main manuscript are displayed between quotation marks in purple font.

Reviewer #1 (Comments to the Author):

In this manuscript, Prindle and colleagues studied the complexes formed by cytosolic components of the type III secretion system in living *Yersinia*. To access the nature, dynamism and requirements of these complexes, they characterized the diffusive states of these proteins, a method they previously established. This work is incremental and now brings new insight into the hetero-oligomeric states of T3SS components prior to interaction with the membrane apparatus. Notably the authors identified a large cytosolic complex including the SctN ATPase, SctL and SctQ that forms independently of the T3SS membrane part. The manuscript is well written and concise. The single-molecule super-resolution fluorescence imaging approach is sound. I only have two main comments:

Major comments:

1. In order to express eYFP fusions with SctL, N and Q as close as possible to native protein levels, the authors engineered chromosomal fusions at the native locus. However, this is no guarantee of success. If for any reason, the stability or production of the fusions is way off compared to native proteins, this could largely affect the populations and dynamic behaviors observed. I would therefore recommend a more careful analysis of fusion protein levels compared to wild-type.

We agree with the Reviewer that having accurate quantification of protein expression levels is desirable. In order to accurately quantify protein expression levels, antibodies against the proteins studied here are needed. Our collaborator on this manuscript (Dr. Diepold) attempted to perform these measurements in his lab. However, the antibodies used were no longer functioning, and so these experiments failed. To our knowledge, the antibody batches in the Diepold lab are the only existing stockpiles against the *Yersinia* injectisome proteins. Raising new antibodies against the *Yersinia* injectisome proteins would be a very costly (>\$10,000) and time-consuming (~1 month) endeavor that beyond the resources available to our labs. Thus, we chose an antibody against GFP for this study. Our Western blot analysis (**Supplementary Fig. 2**) shows slight degradation products for YeSctQ and N, and L does not show any degradation. Importantly, we do not see any evidence of a degradation product in our tracking data (now re-analyzed as described in our response to comment 4 below). Also, expressing the eYFP-labeled proteins in wild-type cells results in close to native levels of effector protein secretion in each case (**Supplementary Fig. 3** and Rocha *et al*, *Integrative Biology*, 2018). Thus, we don't think that expression level difference could be large enough to change the results or conclusions made in this study.

2. Based on the results, the authors proposed that secretion may be regulated by the assembly of cytosolic complexes prior to incorporation into the membrane complex. I really wish the authors went a little bit further to substantiate that compelling idea, unless the experiment required is too complicated to setup. If the hypothesis is correct and reactivation of secretion relies on the formation of cytosolic complexes in permissive conditions, it would be interesting to test if the dissociation of N, L, Q and K from injectisomes, (just after a pH drop as suggested by the authors), releases free or lower order complexes rather than assembled NLQK complexes which could be readily re-incorporated into injectisomes.

We absolutely agree with the Reviewer that pH dependent experiments would be highly illuminating. However, as the Reviewers suggested in comment 4 below, it is critical to measure diffusion under the same environmental conditions. A pH drop would likely also change the viscosity of the cytoplasm. It would therefore be necessary to re-perform tracking of every protein under new pH conditions. Also, eYFP fluorescence is sensitive to pH (Llopis *et al*, PNAS 1998). This pH sensitivity would likely necessitate the use of a different fluorescent label, which would need to be calibrated to test for the possibility of interaction with cytosolic components. For reasons, the pH dependent experiments are outside the scope of this work, but will be addressed in future studies.

Minor comments:

3. The way the fusions are named, ie eYFP-*YeSctQ*, makes it complicated to read, and really, is not useful here.

We agree with the Reviewer and simplified the language in the main text

4. Starting L133 and then L147:
Why don't the authors measure the diffusion rate of eYFP in the same conditions as with eYFP-*YeSctQ* instead of assuming that the diffusion rate of free monomeric eYFP-*YeSctQ* is faster than that of eYFP due to the different experimental conditions?

We thank the Reviewer for suggesting this control experiment. We have performed this experiment and its inclusion in the manuscript has prompted us to re-examine/filter our raw single-molecule localization data more carefully. Including an additional filtering step had the beneficial effect of “cleaning up” our diffusion coefficient spectra, while leaving our previous conclusions unaffected. As a final benefit, the experimental measurement of the eYFP diffusion coefficient in secretion activated cells allowed us to reach a more significant overall conclusion based on the totality of the tracking data reported here.

The (new) findings are:

- Contrary to our previous assumption, the diffusion rate of eYFP actually decreases slightly under secretion-active conditions ($D = 11.9 \rightarrow 9.2 \mu\text{m}^2/\text{s}$). Apparently, the

expected increase in diffusion after the 28°C → 37°C temperature jump is more than offset by the decrease in diffusion that occurs through the addition of salts to the growth medium (osmotic upshift increases biomolecular density in the cytosol) and the addition of glycerol to the growth medium (increasing cytosol viscosity). We have included an entire new section at the very beginning of the results section of the manuscript to quantify the impact of the growth medium on the intracellular diffusion coefficient of eYFP (Lines 107-150). Importantly, the diffusion coefficient spectrum of eYFP is still narrowly peaked, indicating that eYFP is a suitable probe for tracking eYFP-labeled fusion proteins.

- Our previously published YeSctQ tracking results in *Y. enterocolitica* under secretion-active conditions suggested the presence of three diffusive states at $D \sim 1.0$, 4.0 , and $15 \mu\text{m}^2/\text{s}$, with the $D \sim 15 \mu\text{m}^2/\text{s}$ diffusive state constituting a 20% population fraction of all tracked proteins. Such a result is, however, inconsistent with the monomeric eYFP tracking result described above, as an eYFP-labeled protein should not be able to diffuse faster than $9.2 \mu\text{m}^2/\text{s}$ under secretion-active conditions. We therefore re-examined the raw YeSctQ single-molecule trajectories and found that uncharacteristically fast apparent diffusion coefficients are often due to large displacements resulting from mislocalization of fluorescence signals beyond the axial (z-) boundaries of the cell (**Supplementary Fig. 2**). Removal of these localizations and the associated trajectories greatly reduced the fast $D = 15 \mu\text{m}^2/\text{s}$ diffusive states that were previously present in all our datasets. Thus, we no longer observe fast moving states that could be assigned to the monomeric forms of the tracked proteins. Importantly, the diffusive states below $9.2 \mu\text{m}^2/\text{s}$ are largely unaffected by this filtering step.
 - We have edited the manuscript (Lines 158-163 and Supplementary Figure 2) to explain the additional filtering step.
 - We have revised each data figure in the manuscript to now show the results *after* filtering the raw localization data.
- We have revised the final paragraph of the discussion section and included an additional Figure 5 to argue for the following conclusion:

“Previous work by others has shown that the cytosolic diffusion rate of proteins scales with molecular weight, but not according to the Stokes-Einstein equation, which stipulates that $D \propto (\text{Molecular Weight})^{-1/3}$. Instead, a $D \propto (\text{Molecular Weight})^{-2/3}$ scaling has been observed by others independently for different proteins. Here, we obtain a $D \propto (\text{Molecular Weight})^{-0.71}$ scaling when plotting the measured diffusion coefficients against the molecular weights of monomeric eYFP and the protein complexes discussed above (likely stoichiometries were estimated based on current biophysical models of the cytosolic injectisome protein complexes) (**Fig. 5b**). This analysis allows us to distinguish between oligomerization models that differ in terms of protein complex stoichiometry. For example, earlier studies suggested that SctK, Q, Qc, L, and possibly N, form a high molecular weight (>1MDa) complex, termed the sorting platform, that sequentially interacts with different secretion substrates and their chaperones. However, efforts aimed at reconstituting sorting platform complexes *in vitro* have not been reported. The

analysis in Figure 5b suggests that the majority of cytosolic injectisome proteins exist as part of smaller complexes with molecular weights of less than 500 kDa. Our combined results thus narrow down the possible stoichiometries of cytosolic injectisome protein complexes that can form prior to injectisome binding.”

5. L170: Supplementary Figure 2: the legend of the figures, particularly Sup Fig. 2, is not clear and one has to guess what the authors are showing precisely. Molecular mass (numbers) are not indicated in Sup Fig 1. eYFP theoretical mass is 27 kDa and unfortunately, a band is present around that on the gel. In addition, for all three fusions, there are additional bands compared to the WT control (if I understand correctly...). Therefore, I don't see how the authors can rule out the possibility that the peak corresponding to fast diffusing molecules is in fact due to partial cleavage of the fusion.

The band referred to by the Reviewer near 27 kDa was accounted for by performing Western blotting on wild-type *Yersinia enterocolitica*, which serves as our negative control. The bands observed in this lane are consistent across all the other lanes in the blot, indicating that they constitute a background signal due to nonspecific antibody binding. These bands can thus be ignored. Please also refer to our response to Comment 1, where we address the Reviewer's concerns about a cleavage product.

6. L268: are (in) also in close agreement.

We fixed this typo.

7. L309: IN both wild-type and Δ sctQ mutants.

We fixed this typo.

8. L269: Would it be clearer to state that the presence of injectisomes is not necessary to form the cytosolic complexes observed? OR that LN and LNQ complexes form prior to incorporation into injectisomes?.

We agree with the reviewer that the first statement would be helpful. The revised section of manuscript (Lines 300-303) now reads:

“Taken together, these results establish that the same complexes formed in wild-type cells can also form, in increased abundance, in the absence of injectisomes. In other words, the presence of injectisomes is not necessary for cytosolic complex formation.”

9. L321: I don't understand why this argues in favor of a NLQK complex? Doesn't the absence of Q disrupt the NLQ complex, irrespective of the presence of K?.

We thank the Reviewer for catching our imprecise language. We have edited this section to now read:

“Based on these results, we speculate that the $D \sim 1.3 \mu\text{m}^2/\text{s}$ diffusive state we observe in wild-type cells and in ΔsctD mutants is likely not just a SctNLQ complex, but in fact a SctNLQK heterooligomer. This claim is substantiated by the loss of the $D \sim 1.3 \mu\text{m}^2/\text{s}$ diffusive state when we track eYFP-YeSctL and eYFP-YeSctN proteins in the ΔsctQ mutant. Whether a $D \sim 1.3 \mu\text{m}^2/\text{s}$ diffusive state also manifests for SctK in wild-type cells and in ΔsctD mutants remains to be determined.”

Reviewer #2 (Comments to the Author):

The article provides information and proof about the composition and the dynamics of the *Yersinia enterocolitica* sorting platform, including its ATPase SctN. The data indicate the formation of diffusing, cytosolic sorting platform complexes prior to T3SS binding. Overall the paper is sound and it is easy to follow the conclusions the authors drew from the obtained results. Anyhow, it would be important to address the major control experiment mentioned below. In addition, addressing and discussing the minor points would make the manuscript even more comprehensive and could open up discussing points that might be addressed in the future.

Major comment:

1. I suggest to perform an additional control experiment. As already mentioned in lines 147-150, the fitted diffusion rate of $14.8 \mu\text{m}^2/\text{s}$ is faster than the diffusion rate of free eYFP. An additional control experiment under secretion conditions should give further insight if the faster diffusion is really only due to the different conditions. If this is not the case (faster diffusion than free eYFP also under same conditions), please discuss this or potentially use another fitting.

We thank both reviewers for suggesting this important control experiment, since we believe that its inclusion has helped to greatly improve the manuscript.

Please refer to our response to Reviewer 1, Comment 4.

Minor comment:

2. Lines 225-228 and 262-264: Please discuss the reason why there are still slow diffusing molecules ($D < 0.5 \mu\text{m}^2/\text{s}$) in the ΔSctQ and ΔSctD background. In prior experiments this diffusion coefficient was described as stationary/membrane-associated. Here, membrane-associated clusters are absent, but there are still trajectories with $D < 0.5 \mu\text{m}^2/\text{s}$. This is also visible in Fig. 2b compared to 3b and Suppl. Fig. 11.

We thank the Reviewer for pointing out this point of confusion. We classify molecules with slow ($D < 0.5 \mu\text{m}^2/\text{s}$) diffusion coefficients as stationary. However, that classification does not necessarily imply that these molecules are also membrane- or injectisome-bound. To clarify, we have included the following section and an additional supplementary figure (**Supplementary Fig. 1**) in the revised manuscript (Lines 136-141):

“Under secretion-active conditions, we also observe a small (7%) population of slow ($D < 0.5 \mu\text{m}^2/\text{s}$) eYFP, indicating that even small proteins can occasionally get trapped, presumably in crowded pockets of the cytosol (29). We verified that trajectories corresponding to slow moving eYFP are indeed randomly localized to the cytosol and do not show a preference for subcellular locations, such as the membrane or the cell poles (**Supplementary Fig. 1a**). Notably, the slow population is not detectable when tracking eYFP under standard growth conditions.”

Given this observation, the presence of a non-zero population at $D < 0.5 \mu\text{m}^2/\text{s}$ is not unexpected for any fusion protein in the ΔsctQ , ΔsctD , or any other genetic background (see e.g. Fig 3b).

The Figure 5 caption also states that:

“The fitted diffusive states for $D < 0.5 \mu\text{m}^2/\text{s}$ in the ΔsctD , ΔsctQ , and ΔpYV mutants do not localize to the membrane.”

3. Please discuss in more detail the differences between the present study and Rocha et al.,2018. Clarify the importance of the inclusion of SctN in this study.

The current abstract reads:

“The cytosolic injectisome proteins are required for export of effectors and display both stationary, injectisome-bound populations as well as freely-diffusing cytosolic populations. How the cytosolic injectisome proteins interact with each other in the cytosol and associate with membrane-embedded injectisomes remains unclear. Here, we utilize 3D single-molecule tracking to resolve distinct cytosolic complexes of injectisome proteins in living *Yersinia enterocolitica* cells.

SctN is one of four key cytosolic injectisome proteins (others are SctK, Q, and L). The introduction provides a thorough overview of what is currently known about these four proteins. Therefore, we believe the rationale for including SctN in this study is sufficiently clear.

October 20, 2022

Prof. Andreas Gahlmann
University of Virginia
409 McCormick Rd.
Room 212
Charlottesville, Virginia 22904

Re: Spectrum01744-22R1 (Distinct cytosolic complexes containing the type III secretion system ATPase resolved by 3D single-molecule tracking in live *Yersinia enterocolitica*)

Dear Prof. Andreas Gahlmann:

It is my pleasure to inform you that the revised version of your manuscript has been accepted for publication in Microbiology Spectrum, and I am forwarding it to the ASM Journals Department for publication. You will be notified when your proofs are ready to be viewed.

Sincerely,

Giordano Rampioni
Editor, Microbiology Spectrum
